# Comparisons and quality control of wind observations in a mountainous city using wind profile radar and the Aeolus satellite

**Hua Lu[1, 3, 6], Min Xie[2], Wei Zhao[4], Bojun Liu[5, 6], Tijian Wang[1], Bingliang Zhuang[1]**

[1]School of Atmospheric Sciences, Nanjing University, Nanjing 210023, China

[2]School of Environment, Nanjing Normal University, Nanjing 210023, China

[3] Chongqing Institute of Meteorological Sciences, Chongqing 401147, China

[4]Nanjing Institute of Environmental Sciences, Ministry of Ecology and Environment of the People's Republic of China, Nanjing 210023, China

[5]Chongqing Meteorological Observatory, Chongqing 401147, China

[6]Heavy Rain and Drought-Flood Disasters in Plateau and Basin Key Laboratory of Sichuan Province, Chengdu 610072, China

# Comparisons and quality control of wind observations in a mountainous city using wind profile radar and the Aeolus satellite

Hua Lu[1,3,6], Min Xie[2], Wei Zhao[4], Bojun Liu[5,6], Tijian Wang[1], Bingliang Zhuang[1]

[1]School of Atmospheric Sciences, Nanjing University, Nanjing 210023, China

[2]School of Environment, Nanjing Normal University, Nanjing 210023, China

[3] Chongqing Institute of Meteorological Sciences, Chongqing 401147, China

[4] Nanjing Institute of Environmental Sciences, Ministry of Ecology and Environment of the People's Republic of China,Nanjing 210023, China

[5]Chongqing Meteorological Observatory, Chongqing 401147, China

[6]Heavy Rain and Drought-Flood Disasters in Plateau and Basin Key Laboratory of Sichuan Province, Chengdu 610072, China

*Correspondence to*: Min Xie (minxie@njnu.edu.cn) , Wei Zhao (zhaowei@nies.org)

**Abstract:** Observations of vertical wind profile in Chongqing, a typical mountainous city in China, are important, but sparse and have low resolution. To obtain more wind profile data, this study matched the Aeolus track with ground-based wind observation sites in Chongqing in 2021. Based on the obtained results, verification and quality control studies were conducted on the wind observations of a wind profile radar (WPR) with radiosonde (RS) data, and a comparison of the Aeolus Mie-cloudy and Rayleigh-clear wind products with WPR data was then performed. The conclusions can be summarized as follows: (1) A clear correlation between the wind observations of WPR and RS was found, with a correlation coefficient (R) of 0.71. Their root-mean-square deviation increased with height, but decreased at height between 3 and 4 km. (2) After quality control of Gaussian filtering (GF) and empirical orthogonal function construction (EOFc, G = 87.23%) of the WPR data, the R between the WPR and RS reached 0.83 and 0.95, respectively. The vertical distribution showed that GF could better retain the characteristics of WPR wind observations, but with limited improvement in decreasing deviations, whereas EOFc performed better in decreasing deviations, but considerably modified the original characteristics of the wind field, especially regarding intensive vertical wind shear in strong convective weather processes. (3) In terms of the differences between the Aeolus and WPR data, 56.0% and 67.8% deviations were observed within ± 5 m/s for Rayleigh-clear and Mie-cloudy winds vs. WPR winds, respectively. Vertically, large mean differences of both Rayleigh-clean and Mie-cloudy winds versus WPR winds appeared below 1.5 km, which is attributed to the prevailing quiet and small winds within the boundary layer in Chongqing, in this case the

movement of molecules and aerosols is mostly affected by irregular turbulence. Additionally, large
mean differences at the height range between 4 to 8 km for Mie-cloudy versus WPR winds may be
related to the high content of cloud liquid water in the middle troposphere of Chongqing. (4) The
differences in both Rayleigh-clear and Mie-cloudy versus WPR winds had changed. Deviations of
58.9% and 59.6% were concentrated between ±5 m/s for Rayleigh-clear versus WPR winds with GF
and EOFc quality control, respectively. In contrast, 69.1% and 70.2% of deviations appeared between
±5 m/s for Rayleigh-clear versus WPR and EOFc WPR winds, respectively. These results shed light
on the comprehensive applications of multi-source wind profile data in mountainous cities or areas
with sparse ground-based wind observations.
**Keywords:** Wind profile radar, Aeolus satellite, data verification, data quality control, mountainous
city

## 1 Introduction

The detection of the atmospheric wind profile is essential for studying atmospheric dynamics,
interactions between weather and pollution, and predict extreme weather (Baker et al., 1995; King et
al., 2017; Stettner et al., 2019; Sun et al., 2022). Furthermore, the value of atmospheric wind
observations has been illustrated by assimilation applications in numerical weather prediction
(Benjamin et al., 2004; Weissmann and Cardinali, 2007; Michelson and Bao, 2008). In particular,
wind fields within the boundary layer are mostly turbulent and difficult to simulate using models
without the assimilation of wind observations (Belmonte and Stoffelen 2019; Simonin et al., 2014).
For areas with complex terrain, such as mountainous cities, individual ground-based observation
stations usually have poor representation, and thus vertical observations are essential (Sekuła et al.,
2021; Lu et al., 2022b). Therefore, unconventional wind profile observations are urgently required for
analysis and assimilation into numerical prediction models to describe the transport of mesoscale
weather systems, as well as to advance our knowledge of atmospheric component movement in the
actual atmosphere.
Wind profile radar (WPR) data may partially compensate for the limitations of conventional
wind field observations. WPR detects the scattering effect of atmospheric turbulence on
electromagnetic waves to detect the Doppler effect signals of air movement, and is capable of
providing horizontal wind vectors with high temporal and vertical resolution (Weber et al., 1990;

Dibbern et al., 2001). The automated, continuous, and real-time vertical wind profiles from the WPR could fill the gaps in upper-air observations, both in time continuity and vertical resolution. Terrain and climate characteristics in unique regions could have different impacts on WPR echoes, resulting in separate data observation errors. Therefore, data verification, and occasionally adequate quality control, are required before the application of WPR data in a specific region (Zhang et al., 2015; Guo et al., 2020). In comparison, radiosonde (RS) data are often considered reliable atmospheric wind observations to verify WPR data (Weber et al., 1990; Chen et al., 2021).

Owing to advances in satellite detection, wind fields acquired from satellites can supplement conventional ground-based observations in space coverage. Atmospheric motion vector detection can only extract the wind information of layers with clouds. The United States and Europe have successively detected sea surface wind fields using microwave radiometers and scatterometers (Endlich et al., 1971; Njoku et al., 1980; Gaiser et al., 2004; Barre et al., 2008). The World Meteorological Organization regards the detection of global three-dimensional wind fields as one of the most challenging and important meteorological observation missions in the 21st century (WMO, 2001). The United States and Europe have conducted space-borne wind lidar measurement programs, as these are the best methods for detecting three-dimensional wind fields (Beranek et al., 1989; Baker, 2008; Wernham et al., 2016). The Aeolus satellite was launched following the European Space Agency's (ESA) fifth Earth Explorer mission on August 22, 2018. As the world's first Doppler wind lidar in space, Aeolus has enabled the continuous detection of global wind profiles from the ground to the lower stratosphere with a vertical resolution of 0.25–1 km (Marseille et al., 2008; Reitebuch et al., 2006; Zhang et al., 2019). Therefore, the wind profile data detected by Aeolus can compensate for the lack of spatial coverage and vertical resolution of ground-based wind field observations to some extent.

Located at the edge of the Sichuan Basin, Chongqing is a typical mountainous city in China known for its complex topography. Owing to the unique terrain, the mechanism of extreme weather and movement of atmospheric components in the city are intricate and complex, making vertical observations essential. Interference sources for the vertical detection of WPR might form in mountainous areas, which are different from those in plain areas. Thus, reasonable data verification and quality control should be conducted before application to ensure the accuracy and representativeness of the WPR. The spatial distribution of ground-based vertical wind observations in

Chongqing is sparse, and it is worthwhile to verify the performance of Aeolus wind products and
apply them to related mechanistic studies or numerical assimilation systems. To this end, wind profile
observations of RS, WPR, and Aeolus were collected and matched in terms of time and space for
2021 in Chongqing. Based on the matched results, data verification and quality control of WPR wind
observations were implemented using RS data, and the performance of Aeolus wind products in
Chongqing was analyzed to provide a scientific basis for multi-source wind profile data applications
in mountainous cities. The remainder of this paper is organized as follows: the RS, WPR, and Aeolus
data used in this study, the matching procedure, data verification, and quality control methods are
described in Section 2; Section 3 presents the comparison and quality control results of the WPR and
Aeolus wind profile data; finally, the main conclusions are summarized in Section 4.
**2 Data and methods**
**2.1 Data**
**2.1.1 Ground-based wind profile data**
Shapingba (57516; 106.27°E, 29.34°N) is a national weather station and the only RS station in
Chongqing. Wind speed and direction at 0000 and 1200 UTC (universal time coordinated) were
obtained from an L-band sounding system on vertical height levels every 1 s from the surface to 30
km in the air (Zhang et al., 2020). Shapingba station belonged to the network of the L-band sounding
system by China Meteorological Administration. The operational radiosonde stations in China widely
use GTS1 ditital radiosonde as key components of L-band sounding system, which have high
accuracy within the troposphere in detecting fine resolution profiles of meteorological factors (Bian et
al., 2011; Guo et al., 2016; Guo et al.,2021b).
There are two wind profile radars in Chongqing, one at Shapingba station and the other at
Youyang station (57633; 108.76 ° E, 28.84 ° N). Radars can operate almost automatically and
continuously, acquiring vertical profiles of horizontal wind speed and wind direction (Guo et al.,
2021a). The WPR in Shapingba and Youyang are from the same manufacturer, sharing the same
temporal and spatial vertical resolutions of 5 min and 120 m, and vertically detecting 48 and 45 layers
up to 9360 and 8910 m, respectively.
RS wind data are generally reliable vertical observations. Considering Shapingba WPR is
located at the same station with RS, while Youyang Station is 360 km away from the RS, therefore,
the data verification of WPR wind observations was conducted based on Shapingba WPR and RS
data in this study (Figure 1).

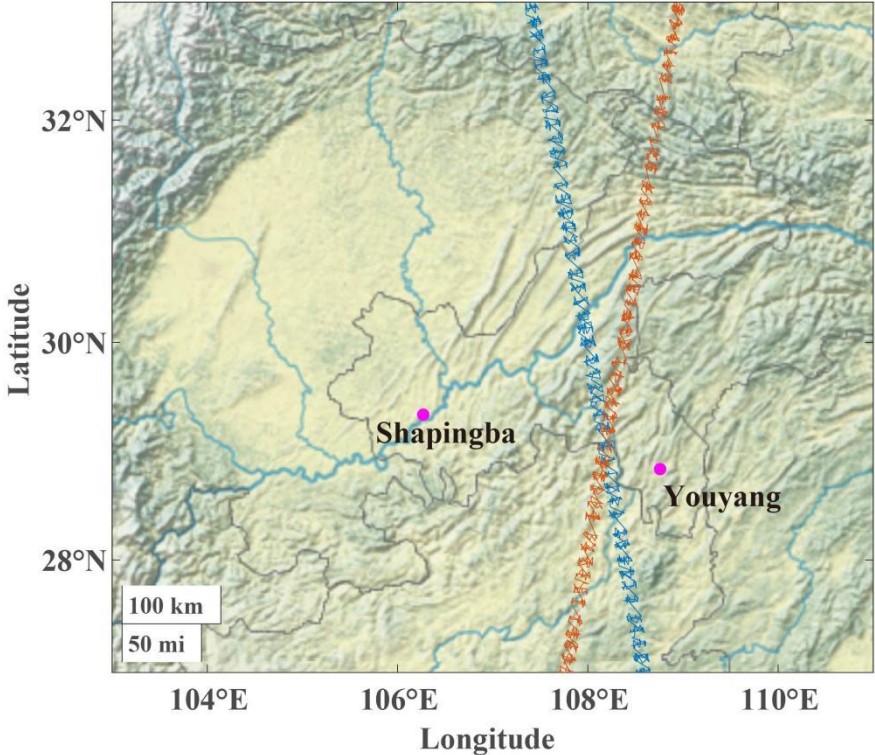


**Figure 1. Geographic locations of ground-based wind observation stations and Aeolus tracks along within**
**Chongqing. The magenta dots denote ground-based observation stations, while red and blue line represent**
**Aeolus trackes. The backgroud is the terrain heights.**
**2.1.2 Aeolus wind products**
Launched on August 22, 2018, the first space-borne Doppler wind lidar, Aeolus, developed by
the ESA, has been circling in a sun-synchronous orbit at an altitude of approximately 320 km, with a
7-day repeat cycle (ESA, 2008). Based on the original detection information, a series of products was
released by the ESA. The Aeolus Level-2B products can provide scientific wind products, which can
be used to obtain wind profile data from the ground to approximately 30 km in the air, with a vertical
resolution of 0.25–2 km and an uncertainty of 2–4 m/s, varying with height (Rennie, 2018; Chen et al.,
2022). Level-2B wind products are classified into Rayleigh-clear and Mie-cloudy winds. Specifically,
Rayleigh channels mainly detect wind fields with atmospheric molecules as tracers in the troposphere
and lower stratosphere, whereas the Mie channel detects signals from aerosols and cloud droplet
particles within the boundary layer or in the cloud (Witschas et al., 2020). In this study, the horizontal
line-of-sight (HLOS) wind products of both Rayleigh and Mie channels were used. Additionally, the
validity flag and estimated errors were extracted for quality control of HLOS wind products (Tan et
al., 2017; Guo et al., 2021a).

**2.2 Methods**


**2.2.1 Data matching and verification procedures**


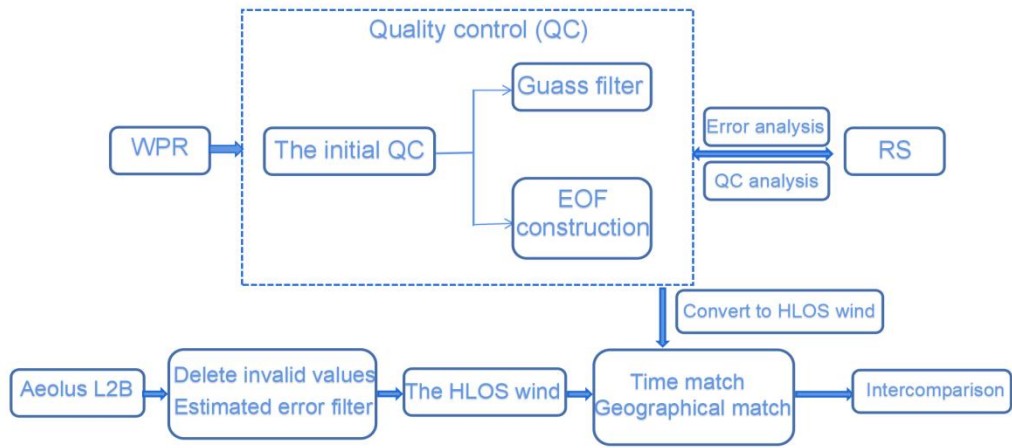

**Figure 2: Flowchart of the multi-source wind profile data matching and verification procedures. WPR stands for wind profile radar, RS stands for radiosonde, EOF stands for empirical orthogonal function.**


In an attempt to make full use of the multi-source vertical wind data from Chongqing,
appropriate procedures were used to match the RS, WPR, and Aeolus data in time and space to make
up the limited ground-based wind profile observations. A flowchart of the procedure is shown in
Figure 2.
First, data verification and quality control effect analysis of the Shapingba WPR were
implemented based on RS data. Based on the approach used by Zhang et al. (2016) and Guo et al.
(2021a), the Aeolus data were removed once the distances between adjacent tracks of Aeolus and
ground-based sites exceeded 1°. With this procedure, Shapingba station is not suitable for comparison
with Aeolus data, whereas Youyang WPR data is. Time and space matches of the WPR and Aeolus
data were posed before the comparison, the geographic location of WPR stations and Aeolus tracks
are shown in Figure 1. Specifically, because of the higher temporal resolution of WPR, the mean
values of WPR data within 10 min before and after Aeolus sampling were used. Vertically, Aeolus
data were interpolated and matched to the layers of WPR data. Subsequently, Aeolus data were
screened by validity flags and estimated errors. Thereafter, both the original Youyang WPR detection
and quality control data were converted into HLOS winds for comparison with the Aeolus data. The
WPR wind vector was projected onto the HLOS winds using the following equation (Witschas et al.,

160  2020):

$$v_{RWP_{HLOS}} = \cos\left(\psi_{Aeolus} - wd_{RWP}\right).ws_{RWP} \qquad (1)$$
where $\psi_{Aeolus}$ is the Aeolus azimuth angle, which could be extracted from the Level 2B products,
while $wd_{RWP}$ and $ws_{RWP}$ are WPR wind direction and speed, respectively.
**2.2.2 Statistical method**
The mean bias (MB) and root mean squared error (RMSE) were adopted as indicators (Equations
2 and 3) for the verification of the WPR and Aeolus wind products, which compares absolute and
relative deviations, respectively.
$$\mathrm{MB} = \frac{1}{n}\sum_{i=1}^{n}\left(o(i) - r(i)\right) \qquad (2)$$
$$\mathrm{RMSE} = \sqrt{\frac{\sum_{i=1}^{n}\left(o(i) - r(i)\right)^2}{n}} \qquad (3)$$
where $o(i)$ represents the observation values and $r(i)$ represents the referent values.
**2.2.3 Data quality control of the wind profile radar**
**2.2.3.1 The initial quality control**
The first step in quality control is to eliminate the abnormal increase of horizontal wind in a
small vertical range of WPR data, including screening invalid data exceeding the climate extreme
values and the vertical consistency test. The extreme climate wind values on the relative layers (Zuo
2020) are listed in Table 1. For the vertical consistency test, if the wind difference between a specific
layer and its adjacent layer is greater than three times that of the two layers below, the value is
considered as an abnormal observation to be deleted (Zhang et al., 2015).
**Table 1: Extreme climate wind values in vertical layers**

| Pressure(hPa) | 1000 | 850 | 700 | 500 | 400 | 300 | 250 |
|---|---|---|---|---|---|---|---|
| Height(m) | 0 | 1500 | 3000 | 5500 | 7000 | 9000 | 10000 |
| Extreme wind(m/s) | 36.01 | 46.30 | 61.73 | 102.89 | 128.61 | 154.33 | 154.33 |

**2.2.3.2 Gaussian filtering (GF) method**

GF is a smooth filtering method that can be used to smooth out the details and noise of two-dimensional graphs, and the observed value of the central point and its surrounding values are summed in one-to-one correspondences. GF is similar to mean filtering, but its preset convolution operator presents a Gaussian distribution. In this study, the convolutional operator was used to calculate the weighted average of the WPR data to filter the high-frequency noise in the observation of WPR. The Gaussian filtering function of the one-dimensional zero-mean normalization is as follows:

$$g(x) = \frac{1}{\sqrt{2\pi}\sigma} e^{\frac{x_2}{2\sigma^2}} \qquad (4)$$

where $\sigma$ is the scale factor that determines the width of the Gaussian filter and further affects the degree of data smoothing. The larger the $\sigma$ value, the wider the frequency band of the Gaussian filter, and the better the data smoothing effect. However, an excessively large $\sigma$ value causes excessive data loss and distortion. In this study, $\sigma$ was set to 3.

**2.2.3.3 Empirical orthogonal function construction (EOFc) method**

Based on the spatial-temporal sequence formed by wind field data W, calculations similar to empirical orthogonal decomposition were performed, and the main modes obtained by calculation were used to reconstruct the spatial-temporal sequence to construct new wind fields. Specifically, the X matrix is formed by selecting N times, a period of time before and after a certain moment, and L layers of effective data, vertically. X is represented below:

$$X = \begin{bmatrix} W_{1,1} & W_{1,2} & \cdots & W_{1,N} \\ W_{2,1} & W_{2,2} & & W_{2,1} \\ \vdots & \vdots & \ddots & \vdots \\ W_{L,1} & W_{L,2} & \cdots & W_{L,N} \end{bmatrix} \qquad (5)$$

Subsequently, the covariance matrix of X, that is, S = XXT, and its eigenvalues and eigenvectors were calculated. According to the arrangement of the eigenvalues from largest to smallest, the cumulative interpretation variance of the first m eigenvectors can be expressed as follows:

$$G = \left( \sum_{k=1}^{m} \lambda_k \right) \Big/ \left( \sum_{k=1}^{L} \lambda_k \right) \qquad (6)$$

The larger the eigenvalue corresponding to the eigenvector, the more its corresponding distribution reflects the typical characteristics of the original field. The time coefficient T = ETX was

calculated with the eigenvector E. Finally, the main modes decomposed by EOF were used to
reconstruct the time series within N times, following the use of X = ET to obtain the vertical
distribution of the wind field at the corresponding time. In the reconstruction of the time series, a
cut-off threshold (G ≥85%) was set for the interpretation of the cumulative variance to control the
quality of the observed data.
Assuming that the cumulative interpretation variances of the first m feature vectors met G ≥85%,
and the first m-1 did not meet G ≥85%, the feature vectors of the first m modes were adopted in the
reconstruction of the sequence, and the corresponding winds at moment j of the ith altitude layer are:
$$WS_{i,j} = \sum_{k=1}^{m} e_{i,k}\, t_{k,j} \qquad (7)$$

The EOFc method can eliminate outliers and pulsating noise from observation data, and has been
applied in quality control research of observational elements in previous studies, such as in Qin et al.

(2010).

### 2.2.4 Quality control of Aeolus wind products

The quality of the Aeolus HLOS wind products is controlled by validity flags and estimated
errors, which are also present in Level 2 B data products. Only data with flags equal to 1 were
considered valid. The data were subsequently filtered according to estimated errors, the theoretical
values calculated based on the measured signal levels, and the temperature and pressure sensitivity of
the Rayleigh channel response (Dabas et al., 2008). Previous studies have revealed that notable
observation errors appeared when the estimated errors were large (Witschas et al., 2020).
Consequently, thresholds for estimated errors of 7(5) m/s were applied for Rayleigh(Mie) winds in
this study, based on the method described by Guo et al. (2021a). Using the parameters valid_flag and
hlos_estimate_error, 18241 Mie-cloudy wind profile samples and 1010 Rayleigh-clear samples were
excluded. As a result, there are 1003 remaining usable Mie-cloudy samples and 1558 remaining
Rayleigh-clear samples. Through the quality control process, significant reductions in the estimated
error were achieved for the Mie-cloudy wind products, from 42.22 m/s to 3.50 m/s. Similarly, for the
Rayleigh-clear wind products, the estimated error has been reduced from 78.69 m/s to 4.58 m/s.

**3 Results and discussion**

**3.1 Data verification and quality control of WPR**

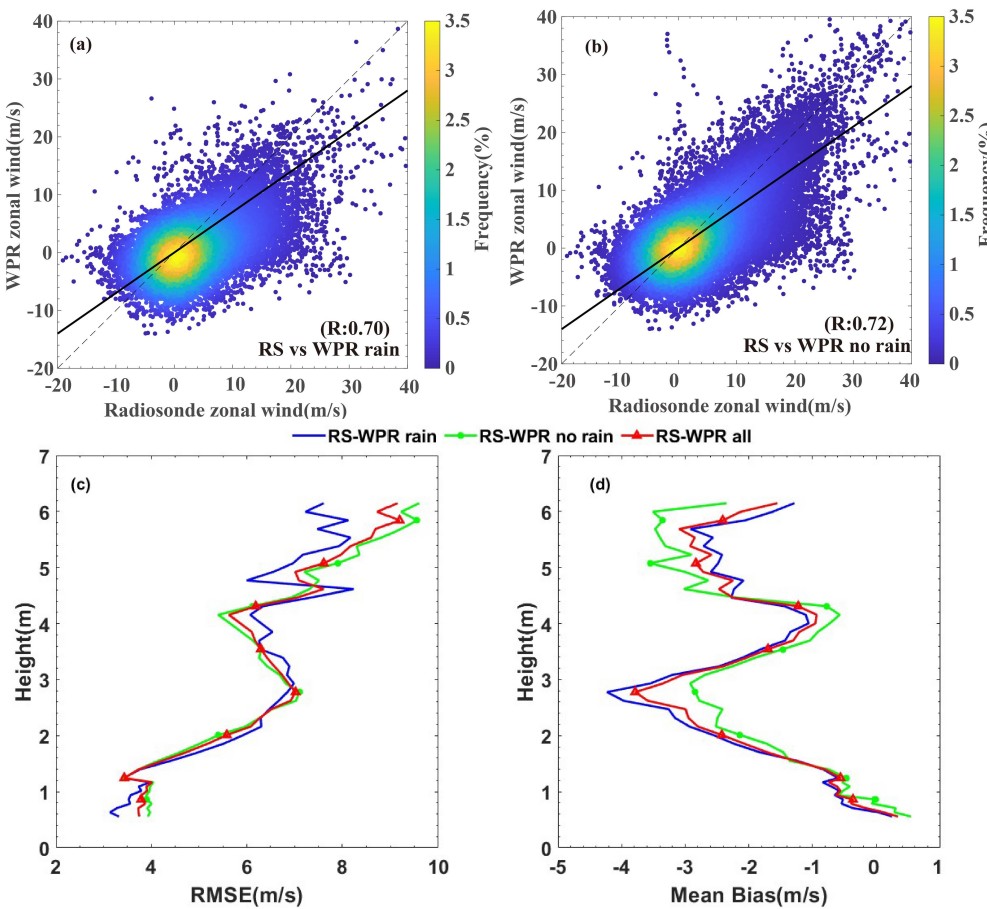

**Figure 3. Scatter density plots for wind profile radar (WPR) vs radiosonde (RS) data during (a) rainy days and (b) no rainy days, and vertical distribution of (c) root mean squared error (RMSE) and (d) mean bias (MB) for WPR vs RS during all days, rainy days and no rainy days.**

Data verification and quality control of the Shapingba WPR were performed based on RS data from the same station. The missing data rate for the Shapingba WPR is 22.78%, resulting in 8117 valid wind profile samples. For the Wulong WPR, the missing data rate is 30.08%, resulting in 7350 valid wind profile samples. RS data has a missing data rate of 13.55%, with 631 valid samples. To address the missing data, different approaches were employed based on the nature of the missing values. When specific levels within a profile have missing data, linear interpolation is used to fill in the gaps. However, if an entire layer of data is missing within a profile, the entire profile is excluded from the analysis.The WPR detects data vertically above the station, while the RS data are derived from air balls, which can respectively drift as far as 0-90, 2-25 and <10 km at 200, 500 and 850 hPa away from the releasing station (Zeng et al., 2019). Therefore, certain differences exist in the spatial

sampling of WPR and RS. Assuming that the atmospheric horizontal distribution is uniform within
dozens of kilometers, the WPR and RS wind fields will be comparable. Additionally, the exact release
times of the air balls were 23:15 and 11:15 UTC, and they generally take 25 min to rise to 10 km.
Therefore, the mean values of the $23:15 - 00:00$ and $11:15 - 12:00$ WPR data were processed to
compare the WPR and RS data. Finally, for comparison with the Aeolus data, wind fields derived
from WPR and RS data were converted into zonal wind components for data verification and quality
control.

To clarify influences of weather, especially precipitation, on wind profile radar observation

quality, scatter plots and vertical distribution of statistical parameters for WPR versus RS during rainy
days and no rainy days were given in Figure 3. Between 1.5 and 4.5 km, WPR deviations during rainy
days exceeded a little that without rain, and the RMSE and MB between WPR and RS were slightly
smaller during rainy days than that without rain below 1.5 km and above 4.5 km. The correlation
coefficient between WPR and RS with rain was a bit lower than that without rain. Generally speaking,
precipitation could affect WPR observation quality, but the deviation distributions were overall the
same during rainy and no rainy days, with slight differences on different layers. As a result, we
discussed the quality control effects of WPR data based on all data, including rainy days and no rain
days.

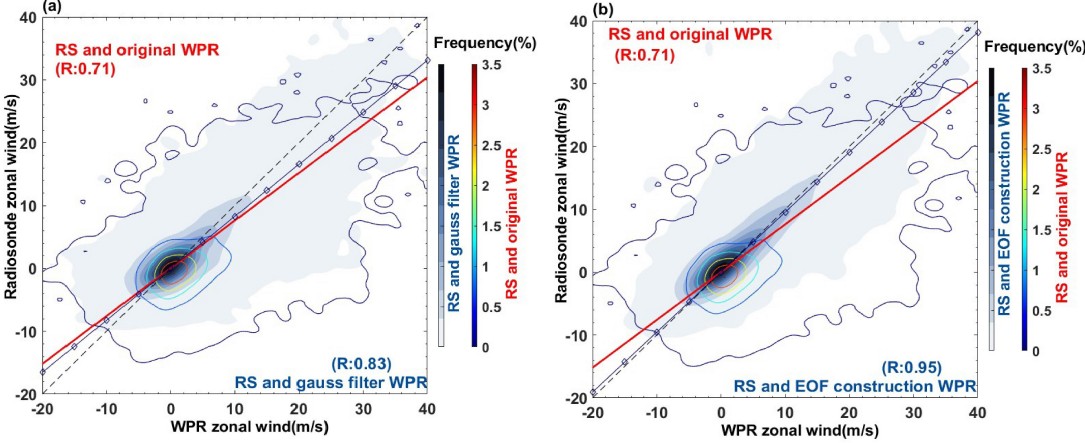


**Figure 4: Scatter density contour plots for (a) original and Gaussian filtering (GF) WPR vs RS data, (b) original and empirical orthogonal function construction (EOFc) WPR vs RS data. In which, the fill contour plots represent original WPR vs RS data, while the contour plots without filling color show GF or EOFc WPR vs RS data.**


Based on Quality Control 1 of the WPR data mentioned above, 784 invalid wind speed data were
filtered, after which GF and EOFc were conducted on WPR winds. The fill contour plots in Figure 4
represent the scatter density distributions of the original WPR and RS data. The correlation
coefficient(R) was 0.71, with scatters distributed along the reference line, indicating a correlation
between the two types of data. Large numbers of dots with significant deviations from the reference
line between the wind speeds of ± 10 m/s implied large differences between the WPR and RS in the
observation of low wind speeds. The contour plots without filling color in Figure 4(a) are scatter
density distributions of GF-controlled WPR and RS, with an R of 0.83, showing better correlation
compared with the original WPR and RS wind data. The GF method screened parts of the data far
away from the reference line, which are wind data with large differences between WPR and RS,
contributing to an improvement in the correlation of the two types of data. The performance of the
WPR data quality control based on EOFc is more significant in Figure 4(b) compared to GF. For
EOFc, G was selected to be greater than 85% for the first time; specifically, the first two modes were
added after EOF decomposition, with G = 87.23%. The R between the EOFc WPR and RS winds
reached 0.95, with scatters more concentrated around the reference line compared with the original
and GF WPR.

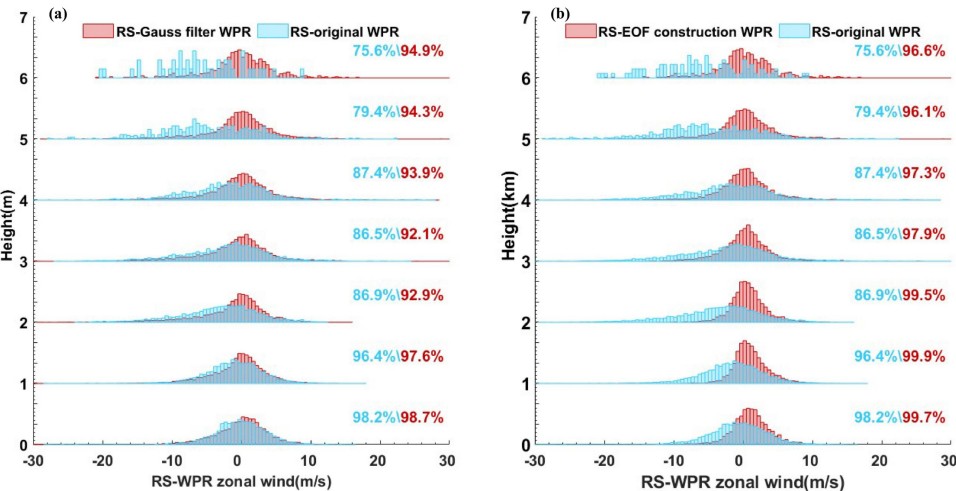


**Figure 5: Probability density distributions vertical variations of (a) RS minus original and GF WPR data, (b) RS minus EOFc WPR data. The blue numbers represent the proportion of RS minus original WPR within -10 to 10 m/s. In (a), the red number represent the proportion of RS minus GF WPR within the range, and in (b), the red for proportion of RS minus EOFc WPR within the range.**


The vertical wind deviation distributions of the original and quality-controlled WPR are shown
in Figure 5, and the vertical distributions of the statistical parameters are shown in Figure 5. The
distribution of deviations between the RS and original WPR data followed normal distribution on
various layers. The median of the distribution was centred around 0 near ground within 2 km, and
gradually moved towards to the negative values above 2 km, indicating significant negative
deviations on the upper layers. Large negative deviations emerged on different layers, however, large
positive deviations mainly distributed around 3-5 km, with the maximum around 30 m/s. Comparing
RS with the original WPR data, 98.2% of the deviations distributed within the -10 to +10 m/s range
near the surface. However, this proportion decreases with increasing altitude, with only 75.6% of the
deviations falling within this range between 6-7 km. Furthermore, when comparing RS with the WPR
data corrected using GF and EOFc, a higher proportion of deviations was observed to concentrate
between -10 to +10 m/s at different altitudes. Specifically, the deviations between RS and EOFc WPR
exhibit a higher proportion of deviations within the -10 to +10 m/s range compared to those between
RS and GF data. From the perspective of statistical parameters, the RMSE of RS and the original
WPR deviation increased with height overall, but decreased at heights between 3 and 4 km. The
vertical MB distribution between the RS and original WPR data presented an M-shaped distribution,
with positive MB values near the ground and negative values in the other layers. According to the
vertical distribution of the deviation scatter points, the negative deviations are significantly larger than
the positive deviations. For a relatively small MB value of approximately 4 km, some of the large
positive deviations in Figure 5 at this level balance the negative values. Similarly, large positive and
negative deviations appeared at approximately 6 km, forming small MB values at this level. In
general, wind speeds increase with height, leading to an increase in the observation deviations of the
WPR.


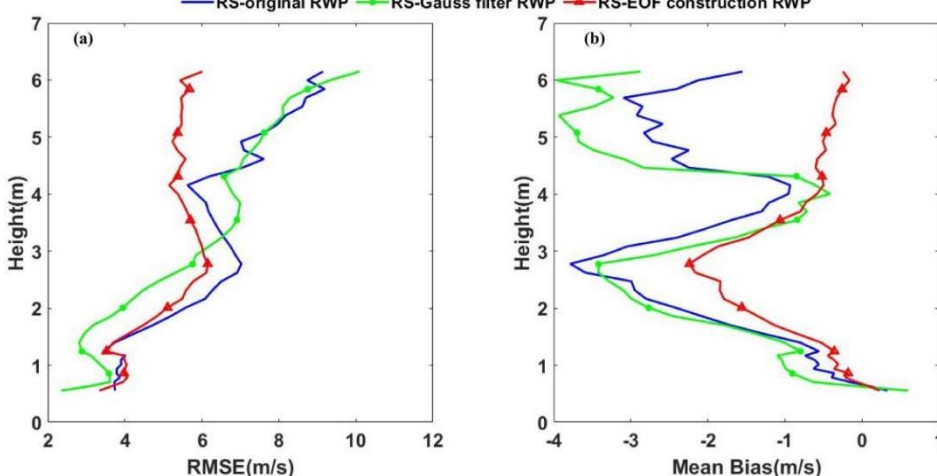


**Figure 6: Vertical distributions of RMSE and MB for (a) RS vs GF WPR data, (b) RS vs EOFc WPR data.**


Taking RS data as true values, the zonal WPR wind data in Chongqing exhibited various
detection errors with height, indicating that quality control of the original WPR data is necessary. The
red histograms in Figure 5(a) represent the vertical deviation distributions between RS data and the
GF WPR with respect to height. Compared with the original WPR data, GF eliminates some large
deviation values of different layers, making the distribution more centred around 0, especially on the
upper layers. The vertical distributions of the RMSE and MB between the RS and WPR data
corresponded to modifications. The RMSE of the RS and GF WPR data is reduced below 3 km
compared to the original WPR, while the alteration of MB mainly manifests above 4 km. Remarkably,
the negative value of MB above 4 km increased after GF in the WPR data. This was because of the
reduction in the larger positive deviation value, and the negative deviation could not be offset.
Subsequently, the EOFc method was adopted for the zonal winds in the original WPR data. The
vertical deviation distributions of RS and EOFc WPR reduced many large negative deviations in the
different vertical layers, making distribution more in line with normal distribution(Figure 5b). The
statistical parameters of the vertical distribution also showed significant changes compared to the
original data. A significant decrease in the RMSE value and a notable reduction in the negative MB
above 1 km were observed between the RS and EOFc WPR (Figure 6). Combining both the vertical
distribution for deviation scatters and statistical parameters, the EOFc WPR winds were similar to the
RS data at various heights. Although the deviations of the two types of data were significantly
reduced, it is worth noting that the EOFc WPR data have modified the characteristics of the original
wind fields to a large extent, especially under strong convective weather conditions with large vertical
wind shear. In comparison, the GF WPR data could better retain the basic characteristics of the
original wind fields. However, the GF method exhibited a limited reduction in the detection
deviations of the WPR data. In general, the two quality control methods have different effects on the
reduction of detection deviations and the retention of the original information.
**3.2 Comparison of the Aeolus and WPR wind data**

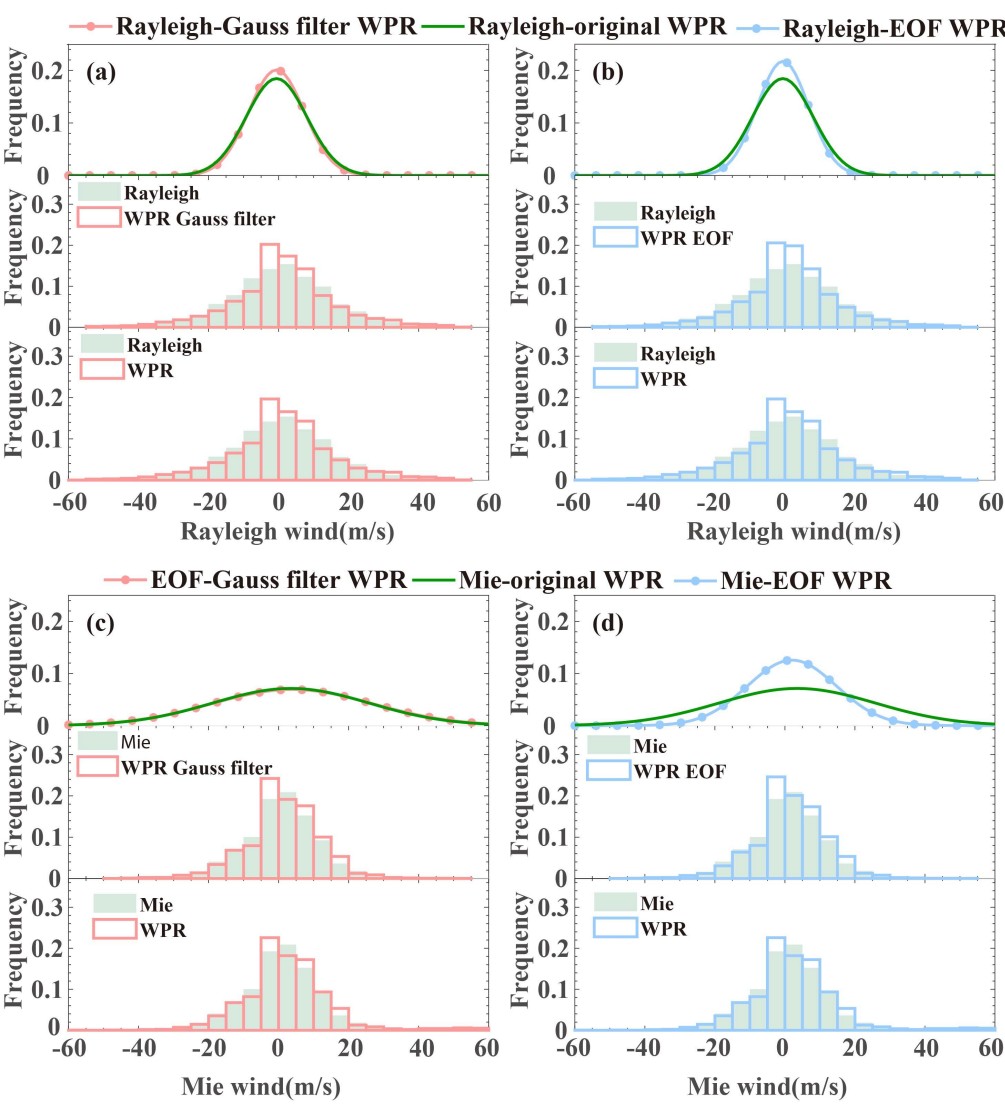


Figure 7: Probability density distributions of deviations and wind distributions of (a) Rayleigh-clear and (c) Mie-cloudy vs WPR original and GF WPR winds, (b) Rayleigh-clear and (d) Mie-cloudy vs original and EOFc WPR winds.


Owing to the limited spatial coverage of ground-based wind profile data, data verification of
Aeolus products in Chongqing was conducted to compensate for the spatial coverage of wind
observations to some extent. The match procedure results indicate that the Youyang WPR data can be
used to verify the Aeolus products described in Section 2. The probability density distribution (PDD)
of deviations between and wind distributions of both Aeolus Rayleigh-clear and Mie-cloudy products
versus WPR data are shown in Figure 7. The PDD of deviations between Rayleigh-clear and WPR in
Figure 7(a) generally present as a Gaussian distribution, with 82.9% of deviations concentrating
between ± 10 m/s and 56.0% of deviations between ± 5 m/s. Quality control with the GF and EOFc
methods was conducted on original WPR observations, and the PDD of deviations between
Rayleigh-clear and quality-controlled WPR winds were concentrated around 0. For deviations
between Rayleigh-clear and GF WPR winds, 85.8% of deviations were centralized between ± 10 m/s
and 58.9% of deviations between ± 5 m/s. In comparison, 86.3% of deviations of Rayleigh-clear and
EOFc WPR winds appeared between ± 10 m/s and 59.6% of deviations between ± 5 m/s. The
scatter distributions of the Rayleigh-clear and WPR winds were shown in Figure 7(a) and 7(b),
respectively. WPR detects winds between -5 and 10 m/s as larger than Rayleigh-clear wind, while it
underestimates wind speeds in the range of ± 10 m/s to ± 20 m/s compared with Aeolus Rayleigh
wind products. Figure 7(c)–(d) show the PDD of deviations and wind distributions of between the
Mie-cloudy and WPR winds. 86.2% of deviations of Mie-cloudy versus original WPR data were
centralized between ±10 m/s and 67.8% of deviations between ±5 m/s, while 86.9% of deviations of
Mie-cloudy versus GF WPR winds were centralized between ± 10 m/s and 69.1% of deviations
between ±5 m/s. For the EOFc WPR winds, 87.5% of deviations appeared between ±10 m/s and
70.2% of deviations between ± 5 m/s. The PDD of wind detected by WPR is similar to that of
Mie-cloudy wind, but WPR generally overestimates wind in the range of -5 and 20 m/s compared
with Aeolus Mie wind products. First, the deviations of the Mie-cloudy and quality-controlled WPR
data were more concentrated around 0 compared with the original WPR.Additionally, compared with
Rayleigh-clear winds, deviations in the Mie-cloudy versus WPR data were small, which may be
attributed to the detection principles of the two channels. Compared with the Rayleigh channel, the
tracers for the Mie channel, including aerosols and cloud droplets within the boundary layer and in
the cloud, mainly centralized at lower vertical levels with smaller wind speeds, resulting in smaller
wind deviations for the Mie-cloudy observations.

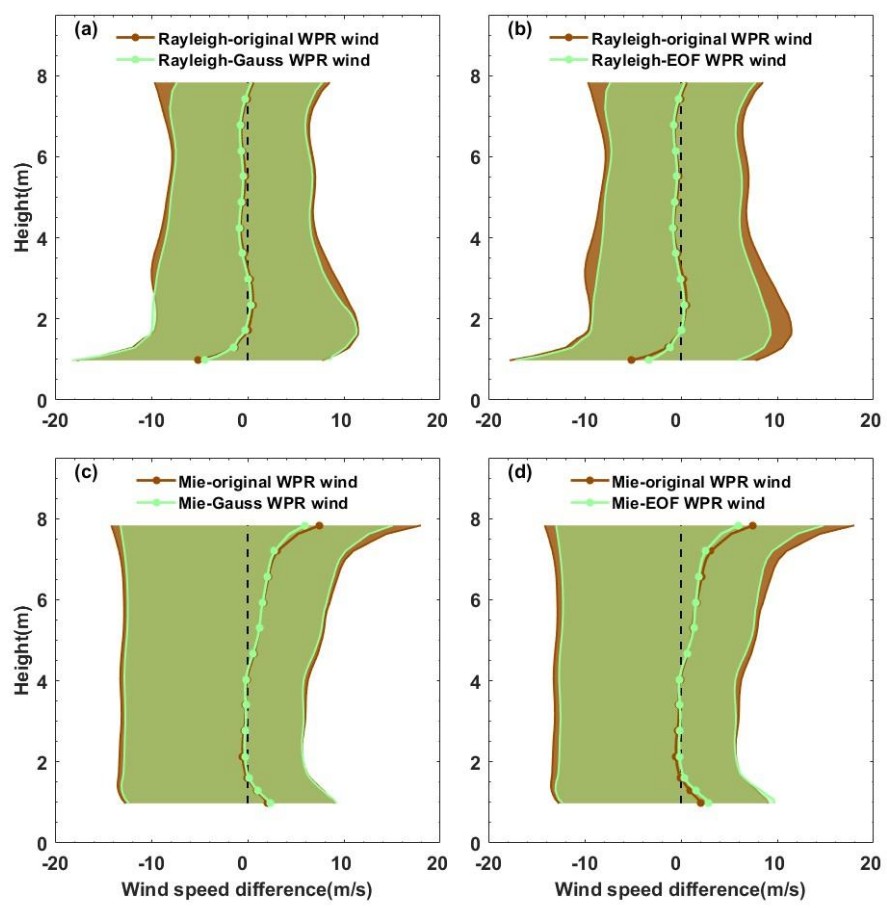


**Figure 8: Vertical distribution of mean differences and deviations between (a) Rayleigh-clear vs GF WPR data, (b) Rayleigh-clear vs original and EOFc WPR data, (c) Mie-cloudy vs original and GF WPR data and (d) Mie-cloudy vs original and EOFc WPR data.**


Figure 8 shows the vertical distribution characteristics of the differences between Aeolus
products and WPR data. The red solid line represents the vertical distributions of the mean differences
between Aeolus and the original WPR data, and the shaded areas denote positive and negative
deviations from the mean differences. Mean differences between the Rayleigh-clear and original
WPR winds have large negative deviations below 1.5 km, with the maximum deviation reaching
-5.2-13.0, -5.2+12.61 m/s. However, the mean difference between these data maintained within ± 1
m/s from the heights of 1.5 to 8 km, with simultaneous decreasing negative and positive deviations
with height. The wind measurement capability of the Rayleigh channel is largely limited by the
receiving intensity, and the Sichuan Basin is one of the large-value aerosol regions in China (Zhang et
al., 2012; Lu et al., 2022a). Particularly, below 1.5 km within the boundary layer, strong aerosol
scattering will inevitably affect molecular scattered signals, thus reducing the accuracy of Rayleigh
channel wind field inversion (Tan et al., 2017; Guo et al., 2021a). In contrast, the vertical distribution
of mean differences between Mie-cloudy and original WPR data (Figure 8c and d) showed large
values within the boundary layer (below 1.5 km) and middle troposphere (4−8 km). The maximal
deviation within the boundary layer reached 2.09-18.23, 2.09+14.76 m/s, while the maximal values
were 7.49-19.98, 7.49+21.64 m/s in the middle troposphere. For the Mie channel, aerosols and cloud
droplet particles were used as tracers for wind measurements. Owing to the influence of the
topography in Chongqing, the prevailing quiet and small winds within the boundary layer result in the
dominant influence of turbulent motion on large particles (Lu et al., 2022b). This contributes to larger
deviations in Mie wind observations because of the irregularity of turbulence. The notable mean
differences in the middle troposphere may be affected by the distribution of cloud droplets. Previous
studies have revealed that due to the influence of the topography of the Tibetan Plateau, the liquid
cloud water contents around 27°N to 35°N in central China are remarkably larger than those in the
southern and northern regions at the same altitude (Yang et al., 2012), with nimbostratus and
altostratus prevailing in the affected areas (Yu et al., 2004). These may contribute to large mean
differences and deviations between Mie winds and WPR data at altitudes of 4−8 km in Chongqing,
which is located on the eastern side of the Tibetan Plateau. According to existing observations, the
frequency of cloud occurrence in the middle troposphere in spring, autumn, and winter is higher than
that in summer, which can explain to some extent why the annual mean differences between Mie
winds and WPR around 4−8 km have large values, whereas the average values in summer do not
(Guo et al., 2021a). Based on the GF and EOFc quality control of the WPR data, the mean differences
between the Rayleigh-clear and WPR winds were found to not change significantly, with only some
reduction in the differences between the Rayleigh-clear and EOFc WPR data within the boundary
layer. However, by controlling the WPR data quality, the positive and negative deviations of the mean
difference at various heights can be effectively reduced (Figure 8a and 6b). Specifically, GF can
reduce deviations above 3 km, whereas EOFc modifies the positive deviations within the boundary
layer. For the Mie winds, a remarkable reduction was observed for mean differences at an altitude of
approximately 6−8 km and deviations in various layers with quality-controlled WPR data compared
with the original WPR data.

## 4 Conclusions

To evaluate the observation quality of the multi-source wind profile data in Chongqing, this study matched the Aeolus, RS, and WPR data for 2021. The matching results indicate that the Youyang WPR can be used for comparison with the Aeolus winds. Additionally, data verification and quality control studies of ground-based WPR data were conducted based on Shapingba RS wind observations. The main conclusions are as follows:

A correlation was found between the RS and original WPR zonal wind data, with an R of 69.92% and scatter points generally distributed along the reference line. The RMSEs of the RS and WPR data increased with height overall, except at an increase of approximately 3–4 km. The MB was vertically distributed in an M-shape, with relatively smaller MB values appearing at 4 and 6 km because of the cancellation of positive and negative deviations.

Screened by the extreme wind climate values and the vertical consistency test, 784 WPR wind observations were eliminated. The R between RS versus GF WPR data and EOFc (G = 87.23) WPR data were 0.83 and 0.95, respectively, demonstrating a better correlation between RS and EOFc WPR data. A comparison of the deviations in the vertical distribution of the RS and WPR data before and after quality control revealed that the EOFc WPR data are closer to RS winds at various heights, resulting in smaller deviations between the two. However, it should be noted that the EOFc WPR winds have a broader filter than the original data, which can remarkably alter the characteristics of the original wind fields, particularly in cases of severe convection weather conditions where there are significant vertical wind shears. While preserving the basic features of the original wind field, the GF method has a limited impact on reducing the deviations of the original WPR wind observations.

The Rayleigh and Mie winds detected by Aeolus exhibited various deviations from the WPR data; 56.0% of deviations between Rayleigh-clear and WPR data existed within ± 5 m/s, while 67.8% of deviations existed between Mie-cloudy and 67.8% of deviations between WPR data were within ± 5 m/s. The Mie channel detects aerosols and cloud droplets as tracers, which are lower than the height layers detected by the Rayleigh channel, resulting in relatively small wind speed deviations. However, the mean differences between Rayleigh-clear and WPR winds are smaller than those of Mie-cloudy winds, especially in the middle troposphere of 4–8 km. This may be due to the influence of the topography of the Tibetan Plateau, resulting in a remarkable increase in the liquid cloud water

content from 27°N to 35°N in central China compared to other regions. Chongqing is located in the
affected areas; thus, the accuracy of Mie wind observations is influenced by the middle troposphere.
The deviations between the Aeolus and WPR data changed to some extent after quality control
of the WPR data, both for the Rayleigh-clear and Mie-cloudy winds. The scatter points of the Aeolus
and WPR data, which were far away from the reference line, decreased; 58.9% of deviations between
the Rayleigh-clear and GF WPR data were centralized between ± 5 m/s, and 59.6% of deviations for
EOFc WPR data were within ± 5 m/s. For the Mie channel, 69.1% of deviations were concentrated
± 5 m/s between the satellite and GF WPR data, and 70.2% of deviations existed between the Mie
and EOFc WPR data. The mean differences of the Rayleigh channel and WPR data changed little
after quality control was conducted using both the GF and EOFc methods on WPR data; however,
both positive and negative deviations to the mean values decreased. For Mie winds, quality control on
WPR made distinct modifications to the mean differences between $6-8$ km and deviations to the
mean values of all layers between Mie-cloudy and WPR data.
**Financial support:** This work was supported by the National Natural Science Foundation of China
(42205186), the Chongqing Natural Science Foundation (cstc2021jcyj-msxmX1007), the open
research fund of Chongqing Meteorological Bureau (KFJJ-201607), Sichuan Science and Technology
Program (2023YFS0430), Heavy Rain and Drought-Flood Disasters in Plateau and Basin Key
Laboratory of Sichuan Province (SZKT202206) and the key technology research and development of
Chongqing Meteorological Bureau (YWJSGG-202215; YWJSGG-202303).
**Acknowledgments:** We would like to express our gratitude to China Meteorological Bureau to provide
the ground-based wind profile data, and the European Space Agency to provide the Aeolus wind
products.
**Conflicts of Interest:** The authors declare no conflict of interest.

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
