# Peer review of "Comparisons and quality control of wind observations in a mountainous city using wind profile radar and the Aeolus satellite"

_Atmospheric Measurement Techniques, 2023_

## Referee Comment (RC2)

**General comments:**

The observation of three-dimensional wind is of importance to weather forecast, air quality and renewable energy. The wind fields in complex terrain like Chongqing are affected by a variety of factors and thus difficulty to be simulated or predicted. Ground- and space-based wind measurements, such as wind profiler radar (WPR) and ALADIN onboard Aeolus, provide an unprecendented opportunity to obtain the vertical profile of wind. Nevertheless, the data quality of Aeolus or WPR in Chongqing remains unknown. The authors conducted comparison analysis used one-year worth of WPR, Aeolus and radiosonde measurements, and revealed some interesting phenomena. The data processing methods, as well as the comparison analysis, are basically scientifically sound. The topic fits in the scope of AMT, and this work is worth of publication in AMT after the authors have fully considered the following comments:

**Major comments:**

1. Apparently, only the wind profile measurements from one WPR station (i.e., Shapingba) is used for verification with Radiosonde observations. If my understanding is right, the comparison analysis between Aeolus and WPR are based on the wind measurements from both WRP stations. But I can not find any descriptions for the WPR at Youyang. For instance, does it have the same frequency? Besides, how far is the WPR station at Youyang from the radiosonde site? the readers are more willing to know the locations of both stations relative to the Aeolus tracks (daytime and nighttime). Therefore, the authors can add one figure in section 2 to illustrate this, and clarify or discuss the potential impact arising from the dismatch of station location.

2. Section 2.1.1 and 2.1.2 can be merged, and "Shapingba WPR is located at the same station as RS; therefore, the data verification of WPR wind observations was conducted based on RS data in this study." can be incorporated to the original section 2.1.2.

3. Figures: The X-axis and Y-axis scale in Figure 2 can be shortened to better show the details of scatters. For example, both axis can be adjusted to -20 to 40 m/s. Figures 2–5: only major ticks in both axes are shown. It is inappropriate for a hig-quality figure not to show the minor ticks.

**Minor comments:**

Line 12: "vertical wind" can be expanded to "vertical wind profile"

Line 19-20: "Their root-mean-square deviation increased with height but decreased by 3 – 4 km." is not clear to me. For example, grammar error exists in "decreased by 3 – 4 km." Besides, I cannot find any figures in this manuscript can support this conclusion, and should be rephrased.

Line 27: "between ± 5 m/s" can be revised to "within ± 5 m/s" or "between -5 m/s to 5 m/s"

Line 28-29: Can you please clarify the specific characteristics in "the mean differences"? ? Otherwise, the sentence "the mean differences… below 1.5 km" makes nonsense.

Line 30: ", such that" can be modified to ". In this case" or similar expression.

Line 31: "large mean differences of 4-8 km" can be rephrased to "Larger mean differences at the height range between 4 to 8 km".

Line 43-44: "to study" -> "for studying…and predicting extreme weather."

Line 33: I wonder the logic behind "influenced by the topography of the Tibetan Plateau." Chongqing (the elevation is less than 4000 m) is in the southwest China, lying far away from the Tibetan Plateau (TP). Besides, do you have any evidence for the connection between cloud liquid water in the middle troposphere and the topography of Chongqing (not TP)? If not, this conjecture can be deleted.

Line 84: "and three-dimensional spatial structure" can be removed.

Line 103: How about the accuracy of the RS measurement in Chongqing, or China? The authors may refer to the following important papers:

https://doi.org/10.1007/s00376-010-9170-8

https://doi.org/10.5194/acp-16-13309-2016

https://doi.org/10.5194/acp-21-17079-2021

Line 104: "Station Shapingba" -> "Shapingba"

Line 156 and 174: "Where" should be revised to "where" and no indent before "where"

Line 213: "which drift more than 10 kilometers away from the releasing station", the drifting distrance of RS balloone depends on the altitude, and this expression can be revised by referring to Figure 2 in Zeng et al. JGR 2019 (https:// doi.org/10.1029/2018JD029109)
Line 67: Is the publication year in "Zhang et al. 2017" supposed to be 2015?

---

## Author Comment (AC1)

Dear Editors and Reviewer:

Thank you very much for your careful review and valuable suggestions with regard to our manuscript "Comparisons and quality control of wind observations in a mountainous city using wind profile radar and the Aeolus satellite" (Manuscript Number: amt-2023-152). The comments are helpful for revising and improving our paper. We have carefully studied these comments and made changes in the manuscript according to reviewers' comments.

**General comments**

This study conducted data verification and quality control on wind profile radar and Aeolus wind products, trying to enrich the available wind observations in regions with complex regions. This kind of study is needed to provide more reliable wind observations for both related mechanistic studies and assimilation applications in numerical weather prediction. Generally, this work is quite meaningful and informative. Most results are pretty valuable to atmospheric measurement studies. I would recommend its acceptance for publication after some necessary revisions.

Response: We would like to thank the reviewer for the valuable and affirmative comments of our manuscript.

**Specific comments**

1. Line 85: "determining the movement of atmospheric components", what determining the movement of atmospheric components? Please reorganize this sentence.

Response: Thanks for the careful suggestion. To make it clearer, we have reorganized this sentence as: "Owing to the unique terrain, the mechanism of extreme weather and movement of atmospheric components are intricate and complex".

2. In "2.1 Data", the location of wind profile radar, radiosonde station, and Aeolus tracks used in this study should be showed in a figure.

Response: Thanks for this suggestion. We have added Figure 1 to show the location of ground-based observation stations and Aeolus tracks, and rearranged the order of Figure 2-8.

[Figure]

Figure 1. Geographic locations of ground-based wind observation stations and Aeolus tracks along within Chongqing. The magenta dots denote ground-based observation stations, while red and blue line represent Aeolus trackes. The backgroud is the terrain heights.

3. Line 112: "Radar can operate almost automatically" should be "Radars can operate almost automatically", as there are two radars.

Response: We are sorry for this clerical error and have made modification in the revised manuscript.

4. In "2.2 Methods", The number labels of equations are missing in this manuscript. Please add the labels, so that the readers could find corresponding equations.

Response: We are sorry for the neglect and have added labels for Equation 1-7 in "2.2 Methods" of the revised manuscript.

5. In Table 1: how do the authors get these extreme climate wind values?

Response: Thanks for this comment. We get these extreme climate wind values referring to Zuo(2020). The detail of the reference is as below:

Zuo Q. M.S. 2020. Research on Quality Control Methods and Assimilation Application of Wind Profiler Radar Data. Nanjing: Nanjing University of Information Science and Technology.

We have added the citing when first mentioning the table and the paper in the Reference part of the revised manuscript.

6. The resolutions of figures in this manuscript should be improved, especially for the label and legends.

Response: We appreciate this suggestion and feel sorry for the inconvenience in reading. We have re-plotted the figures in the revised manuscript.

7. In Figure 2, the red scatter plots and blue ones overlap to a great extent, not very clearly expressing relationships between data. The readers may want some objective statistical data on the figure, like the correlation coefficient, which could be more intuitive to illustrate the relationships.

Response: We are appreciated for this comment. We agree that the red scatter plots and blue ones overlap to a great extent that does not clearly express relationships between data. To show readers the objective statistical data of the figure and make it more intuitive to illustrate the relationships, we have added correlation coefficient in both sub-figures.

[Figure]

**Figure 4: Scatter-plots for (a) original and Gaussian filtering (GF) WPR vs RS data, (b) original and empirical orthogonal function construction (EOFc) WPR vs RS data.**

8. In 3.1, the authors should provide more in-depth analysis for data verification during different weather conditions, because as far as we know, wind profile radar observations may be influenced largely by the weather, rainy or not.

Response: Thanks for the valuable comment. To clarify influences of weather on wind profile radar observation quality, we added Figure 3 that includes scatter plots and vertical distribution of statistical parameters for WPR versus RS during rainy days and no rainy days. Between 1.5 and 4.5 km, WPR deviations during rainy days exceeded a little that without rain, and the RMSE and MB between WPR and RS were slightly smaller during rainy days than that without rain below

1.5km and above 4.5km. The correlation coefficient between WPR and RS with rain was a bit lower than that without rain. Generally speaking, precipitation could affect WPR observation quality, but the deviation distributions were overall the same during rainy and no rainy days, with slight differences on different layers. For details, please see Figure 3 and the corresponding descriptions in Line 242-251 of the manuscript.

[Figure]

**Figure 3. Scatter-plots for wind profile radar (WPR) vs radiosonde (RS) data during (a) rainy days and (b) no rainy days, and vertical distribution of (c) root mean squared error (RMSE) and (d) mean bias (MB) for WPR vs RS during all days, rainy days and no rainy days.**

9. Line 213: "which drift more than 10 kilometers away from the releasing station", the RS air balls may not always drift more than 10 kilometers away, but in the high levels with large winds, please modify the expression to make it more suitable.

Response: Thanks for this rigorous comment. We have modified the expression referring to Zeng et al. (2019) as "which can respectively drift as far as 0-90, 2-25 and <10km at 200, 500 and 850hPa away from the releasing station (Zeng et al., 2019)", and added this paper for Reference in the revised manuscript.

10. Line 227: "The number of dots ...."? The authors might want to say "large numbers of dot ...". Please make correction.

Response: We are sorry for this mistake, and have modified the sentence as "large numbers of dots..." in the revised manuscript.

11. Line 324-325: "at a height of 1km, the mean difference between these data was maintained within ± 1 m/s", but it showed large negative deviations below 1.5 km in the figure.

Response: We are appreciated for this comment. It should be "the mean difference between these data maintained within ± 1 m/s from the heights of 1.5 to 8km".

12. The paper has some strange expressions and grammatical mistakes in writing, which should be corrected. For example, there are some mix uses of tense. On lines 211-213, the first sentence is past tense, but the second sentence is present tense. Please check throughout the manuscript about this problem.

Response: Thanks for this valuable comments. We used past tense as the first sentence described a specific action, while present tense for the second sentence when it described an objective fact. However, there are some other mistakes in tenses. We have checked throughout the manuscript about this problem and made modification.

Sincerely,

Authors

---

## Author Comment (AC2)

Dear Editors and Reviewer:

Thank you very much for your careful review and valuable suggestions with regard to our manuscript "Comparisons and quality control of wind observations in a mountainous city using wind profile radar and the Aeolus satellite" (Manuscript Number: amt-2023-152). The comments are helpful for revising and improving our paper. We have carefully studied these comments and made changes in the manuscript according to reviewer's comments. The main corrections in the manuscript and responses to the reviewer' comments are listed as follows.

**General comments:**

The observation of three-dimensional wind is of importance to weather forecast, air quality and renewable energy. The wind fields in complex terrain like Chongqing are affected by a variety of factors and thus difficulty to be simulated or predicted. Ground- and space-based wind measurements, such as wind profiler radar (WPR) and ALADIN onboard Aeolus, provide an unprecedented opportunity to obtain the vertical profile of wind. Nevertheless, the data quality of Aeolus or WPR in Chongqing remains unknown. The authors conducted comparison analysis used one-year worth of WPR, Aeolus and radiosonde measurements, and revealed some interesting phenomena. The data processing methods, as well as the comparison analysis, are basically scientifically sound. The topic fits in the scope of AMT, and this work is worth of publication in AMT after the authors have fully considered the following comments:

Response: We earnestly appreciate for the reviewer's warm work, and have made modifications according to the valuable comments. The details are as below.

**Major comments:**

1. Apparently, only the wind profile measurements from one WPR station (i.e., Shapingba) is used for verification with Radiosonde observations. If my understanding is right, the comparison analysis between Aeolus and WPR are based on the wind measurements from both WRP stations. But I can not find any descriptions for the WPR at Youyang. For instance, does it have the same frequency? Besides, how far is the WPR station at Youyang from the radiosonde site? the readers are more willing to know the locations of both stations relative to the Aeolus tracks (daytime and nighttime). Therefore, the authors can add one figure in section 2 to illustrate this, and clarify or discuss the potential impact arising from the dismatch of station location.

Response: Thanks for this valuable comment.

First of all, the comparison analysis between Aeolus and WPR is based on Youyang station. According to studies of Zhang et al. (2016) and Guo et al. (2021), Shapingba Station was excluded for comparison, because its distances to adjacent tracks of Aeolus exceeded 1°. The detail

description was given in the revised manuscript as "First, data verification and quality control effect analysis of the Shapingba WPR were implemented based on RS data. Based on the approach used by Zhang et al. (2016) and Guo et al. (2021a), the Aeolus data were removed once the distances between adjacent tracks of Aeolus and ground-based sites exceeded 1°. ".

Secondly, for the description of WPR at Youyang, we have followed the suggestion and added some detail description in Section 2.1.1 Ground-based wind profile data. The WPR in Youyang share same temporal and spatial vertical resolution of 5 min and 120 m with Shapingba. And the distance of Youyang from radiosonde site in Shapingba is more than 360 km. The detailed description and figure in the revised manuscript is as below:

There are two wind profile radars in Chongqing, one at Shapingba station and the other at Youyang station (57633; 108.76°E, 28.84°N). Radars can operate almost automatically and continuously, acquiring vertical profiles of horizontal wind speed and wind direction (Guo et al., 2021a). The WPR in Shapingba and Youyang are from the same manufacturer, sharing the same temporal and spatial vertical resolutions of 5 min and 120 m, and vertically detecting 48 and 45 layers up to 9360 and 8910 m, respectively.

RS wind data are generally reliable vertical observations. Considering Shapingba WPR is located at the same station with RS, while Youyang Station is 360 km away from the RS, therefore, the data verification of WPR wind observations was conducted based on Shapingba WPR and RS data in this study (Figure 1).

[Figure]

**Figure 1. Geographic locations of ground-based wind observation stations and Aeolus tracks along within Chongqing. The magenta dots denote ground-based observation stations, while red and blue line represent Aeolus trackes. The backgroud is the terrain heights.**

Finally, we have added Figure 1 to illustrate the locations of both ground-based stations and the Aeolus tracks, with Figure 1 in the revised manuscript. The discussion about the potential impact arising from the dismatch of station location was given in the revised manuscript as below:

"data verification and quality control effect analysis of the Shapingba WPR were implemented based on RS data. Based on the approach used by Zhang et al. (2016) and Guo et al. (2021a), the Aeolus data were removed once the distances between adjacent tracks of Aeolus and ground-based sites exceeded 1°. With this procedure, Shapingba station is not suitable for comparison with Aeolus data, whereas Youyang WPR data is. Time and space matches of the WPR and Aeolus data were posed before the comparison, the geographic location of WRP stations and Aeolus tracks are shown in Figure 1. "

2. Section 2.1.1 and 2.1.2 can be merged, and "Shapingba WPR is located at the same station as RS; therefore, the data verification of WPR wind observations was conducted based on RS data in this study." can be incorporated to the original section 2.1.2.

Response: We are appreciated for this comment and have followed this suggestion. The original Section 2.1.1 and 2.1.2 have been merged to be Section 2.1.1 Ground-based wind profile data. The

description in Section 2.1.1 in the revised manuscript has also been rephrased.

3. Figures: The X-axis and Y-axis scale in Figure 2 can be shortened to better show the details of scatters. For example, both axis can be adjusted to -20 to 40 m/s. Figures 2–5: only major ticks in both axes are shown. It is inappropriate for a high-quality figure not to show the minor ticks.

Response: We are thankful for the careful comment. Following this suggestion, we have adjusted the axis in original Figure 2 and added minor ticks in other figures.

**Minor comments:**

Line 12: "vertical wind" can be expanded to "vertical wind profile"

Response: Thanks for this kind remind. We have modified "vertical wind" to "vertical wind profile" in the revised manuscript.

Line 19-20: "Their root-mean-square deviation increased with height but decreased by 3 – 4 km." is not clear to me. For example, grammar error exists in "decreased by 3 – 4 km." Besides, I cannot find any figures in this manuscript can support this conclusion, and should be rephrased.

Response: We feel sorry for the unclear expression, and have rephrased this sentence as "Their root-mean-square deviation increased with height, but decreased at heights between 3 and 4 km." This conclusion is derived from Figure 6(a) in the revised manuscript, with the blue line illustrating vertical variation of RMSE between RS and the WPR. Please see Figure 6 and the corresponding description:

[Figure]

Figure 6: Vertical distributions of RMSE and MB for (a) RS vs GF WPR data, (b) RS vs EOFc WPR data.

Large negative deviations emerged on different layers, however, large positive deviations were mainly distributed around 3-5 km, with the maximum around 30 m/s. From the perspective of statistical parameters, the RMSE of RS and the original WPR deviation increased with height overall, but decreased at heights between 3 and 4 km.

Line 27: "between ± 5 m/s" can be revised to "within ± 5 m/s" or "between -5 m/s to 5 m/s"

Response: Thanks for this kind comment. We have revised the expression as "within ± 5 m/s" in the revised manuscript.

Line 28-29: Can you please clarify the specific characteristics in "the mean differences"?? Otherwise, the sentence "the mean differences… below 1.5 km" makes nonsense.

Response: We are sorry for the lack of clarity and have rephrased the expression as "large mean differences of both Rayleigh-clean and Mie-cloudy winds versus WPR winds appeared below 1.5km".

Line 30: ", such that" can be modified to ". In this case" or similar expression.

Response: Thank you for the comment. We have followed it in the revised manuscript.

Line 31: "large mean differences of 4-8 km" can be rephrased to "Larger mean differences at the height range between 4 to 8 km".

Response: We are thankful for this comment, and have rephrased the sentence following the suggestion in the revised manuscript.

Line 43-44: "to study" -> "for studying…and predicting extreme weather."

Response: Thanks for the careful comment. We have modified "to study" to "for studying" in the revised manuscript.

Line 33: I wonder the logic behind "influenced by the topography of the Tibetan Plateau." Chongqing (the elevation is less than 4000 m) is in the southwest China, lying far away from the Tibetan Plateau (TP). Besides, do you have any evidence for the connection between cloud liquid water in the middle troposphere and the topography of Chongqing (not TP)? If not, this conjecture can be deleted.

Response: We are appreciated for the rigorous suggestion and have deleted this expression in the revised manuscript.

Line 84: "and three-dimensional spatial structure" can be removed.

Response: We are thankful for the suggestion and have followed it the revised manuscript.

Line 103: How about the accuracy of the RS measurement in Chongqing, or China? The authors

may refer to the following important papers:

https://doi.org/10.1007/s00376-010-9170-8

https://doi.org/10.5194/acp-16-13309-2016

https://doi.org/10.5194/acp-21-17079-2021

Response: Thank you for the recommendations. Referring to these papers, we have added some description about accuracy of RS measurement in China as:

"Shapingba station belonged to the network of the L-band sounding system by China Meteorological Administration. The operational radiosonde stations in China widely use GTS1 digital radiosonde as key components of L-band sounding system, which have high accuracy within the troposphere in detecting fine resolution profiles of meteorological factors (Bian et al., 2011; Guo et al., 2016; Guo et al.,2021)."

The corresponding references were also added in Reference.

Line 104: "Station Shapingba" -> "Shapingba"

Response: Thanks for the careful suggestion. We have made modification in the revised manuscript.

Line 156 and 174: "Where" should be revised to "where" and no indent before "where"

Response: We are sorry for the format errors and have corrected this kind of errors in the revised manuscript.

Line 213: "which drift more than 10 kilometers away from the releasing station", the drifting distrance of RS balloone depends on the altitude, and this expression can be revised by referring to Figure 2 in Zeng et al. JGR 2019 (https:// doi.org/10.1029/2018JD029109)

Response: Thanks for the recommendation. We have modified the expression referring to Zeng et al. (2019) as "which can respectively drift as far as 0-90, 2-25 and <10km at 200, 500 and 850hPa away from the releasing station (Zeng et al., 2019)" in the revised manuscript.

Line 67: Is the publication year in "Zhang et al. 2017" supposed to be 2015?

Response: Thanks for the kind remind. We have checked the publication year of the reference and correct it in the revised manuscript.

Sincerely,

Authors

---

## Author Response (AR1)

Dear Editors and Reviewers:

Thank you very much for your careful review and valuable suggestions with regard to our manuscript "Comparisons and quality control of wind observations in a mountainous city using wind profile radar and the Aeolus satellite" (Manuscript Number: amt-2023-152). The comments are helpful for revising and improving our paper. We have carefully studied these comments and made changes in the manuscript according to reviewers' comments. The main corrections in the manuscript and responses to the reviewer' comments are listed as follows, and also shown in the pdf file "amt-2023-152-revised-manuscript.pdf". Besides, revision records can be found in the pdf file with revisions mode named "amt-2023-152-revised_manuscript_with_marks.pdf".

Reviewer #1:

**General comments**

This study conducted data verification and quality control on wind profile radar and Aeolus wind products, trying to enrich the available wind observations in regions with complex regions. This kind of study is needed to provide more reliable wind observations for both related mechanistic studies and assimilation applications in numerical weather prediction. Generally, this work is quite meaningful and informative. Most results are pretty valuable to atmospheric measurement studies. I would recommend its acceptance for publication after some necessary revisions.

Response: We would like to thank the reviewer for the valuable and affirmative comments of our manuscript.

**Specific comments**

1. Line 85: "determining the movement of atmospheric components", what determining the movement of atmospheric components? Please reorganize this sentence.

Response: Thanks for the careful suggestion. To make it clearer, we have reorganized this sentence as: "Owing to the unique terrain, the mechanism of extreme weather and movement of atmospheric components are intricate and complex". Please see Lines 84-85 in the revised manuscript.

2. In "2.1 Data", the location of wind profile radar, radiosonde station, and Aeolus tracks used in this study should be showed in a figure.

Response: Thanks for this suggestion. We have added Figure 1 to show the location of ground-based observation stations and Aeolus tracks, and rearranged the order of Figure 2-8. Please see figures in the revised manuscript.

3. Line 112: "Radar can operate almost automatically" should be "Radars can operate almost automatically", as there are two radars.

Response: We are sorry for this clerical error and have made modification in Line 112 in the revised manuscript.

4. In "2.2 Methods", The number labels of equations are missing in this manuscript. Please add the labels, so that the readers could find corresponding equations.

Response: We are sorry for the neglect and have added labels for Equation 1-7 in "2.2 Methods" of the revised manuscript.

5. In Table 1: how do the authors get these extreme climate wind values?

Response: Thanks for this comment. We get these extreme climate wind values referring to Zuo(2020). The detail of the reference is as below:

Zuo Q. M.S. 2020. Research on Quality Control Methods and Assimilation Application of Wind Profiler Radar Data. Nanjing: Nanjing University of Information Science and Technology.

We have added this reference in Line 173-174 when first mentioning the table and in Line 567-568 of the Reference part in the revised manuscript.

6. The resolutions of figures in this manuscript should be improved, especially for the label and legends.

Response: We appreciate this suggestion and feel sorry for the inconvenience in reading. We have re-plotted the figures in the revised manuscript. Please see figures in the revised manuscript.

7. In Figure 2, the red scatter plots and blue ones overlap to a great extent, not very clearly expressing relationships between data. The readers may want some objective statistical data on the figure, like the correlation coefficient, which could be more intuitive to illustrate the relationships.

Response: We are appreciated for this comment. We agree that the red scatter plots and blue ones overlap to a great extent that does not clearly express relationships between data. To show readers the objective statistical data of the figure and make it more intuitive to illustrate the relationships, we have added correlation coefficient in both sub-figures. Please see Figure 4 in the revised manuscript.

8. In 3.1, the authors should provide more in-depth analysis for data verification during different weather conditions, because as far as we know, wind profile radar observations may be influenced

largely by the weather, rainy or not.

Response: Thanks for the valuable comment. To clarify influences of weather on wind profile radar observation quality, we added Figure 3 that includes scatter plots and vertical distribution of statistical parameters for WPR versus RS during rainy days and no rainy days. Between 1.5 and 4.5 km, WPR deviations during rainy days exceeded a little that without rain, and the RMSE and MB between WPR and RS were slightly smaller during rainy days than that without rain below 1.5km and above 4.5km. The correlation coefficient between WPR and RS with rain was a bit lower than that without rain. Generally speaking, precipitation could affect WPR observation quality, but the deviation distributions were overall the same during rainy and no rainy days, with slight differences on different layers. For details, please see Figure 3 and the corresponding descriptions in Line 242-251 of the manuscript.

9. Line 213: "which drift more than 10 kilometers away from the releasing station", the RS air balls may not always drift more than 10 kilometers away, but in the high levels with large winds, please modify the expression to make it more suitable.

Response: Thanks for this rigorous comment. We have modified the expression referring to Zeng et al. (2019) as "which can respectively drift as far as 0-90, 2-25 and <10km at 200, 500 and 850hPa away from the releasing station (Zeng et al., 2019)". Please see Line 233-234 for corresponding description and Line 549-550 for Reference in the revised manuscript.

10. Line 227: "The number of dots ...."? The authors might want to say "large numbers of dot ...". Please make correction.

Response: We are sorry for this mistake, and have modified the sentence as "large numbers of dots..." in Line 258 of the revised manuscript.

11. Line 324-325: "at a height of 1km, the mean difference between these data was maintained within ± 1 m/s", but it showed large negative deviations below 1.5 km in the figure.

Response: We are appreciated for this comment. It should be "the mean difference between these data maintained within ± 1 m/s from the heights of 1.5 to 8km". Please see Line 355-356 of the revised manuscript for corresponding modification.

12. The paper has some strange expressions and grammatical mistakes in writing, which should be corrected. For example, there are some mix uses of tense. On lines 211-213, the first sentence is past tense, but the second sentence is present tense. Please check throughout the manuscript about

this problem.

Response: Thanks for this valuable comments. We used past tense as the first sentence described a specific action, while present tense for the second sentence when it described an objective fact. However, there are some other mistakes in tenses. We have checked throughout the manuscript about this problem. Please see Line 237, 259, 265-266, 280, 295 and 329 of the revised manuscript.

Reviewer #2:

**General comments:**

The observation of three-dimensional wind is of importance to weather forecast, air quality and renewable energy. The wind fields in complex terrain like Chongqing are affected by a variety of factors and thus difficulty to be simulated or predicted. Ground- and space-based wind measurements, such as wind profiler radar (WPR) and ALADIN onboard Aeolus, provide an unprecendented opportunity to obtain the vertical profile of wind. Nevertheless, the data quality of Aeolus or WPR in Chongqing remains unknown. The authors conducted comparison analysis used one-year worth of WPR, Aeolus and radiosonde measurements, and revealed some interesting phenomena. The data processing methods, as well as the comparison analysis, are basically scientifically sound. The topic fits in the scope of AMT, and this work is worth of publication in AMT after the authors have fully considered the following comments:

Response: We earnestly appreciate for the reviewer's warm work, and have made modifications according to the valuable comments. The details are as below.

**Major comments:**

1. Apparently, only the wind profile measurements from one WPR station (i.e., Shapingba) is used for verification with Radiosonde observations. If my understanding is right, the comparison analysis between Aeolus and WPR are based on the wind measurements from both WRP stations. But I can not find any descriptions for the WPR at Youyang. For instance, does it have the same frequency? Besides, how far is the WPR station at Youyang from the radiosonde site? the readers are more willing to know the locations of both stations relative to the Aeolus tracks (daytime and nighttime). Therefore, the authors can add one figure in section 2 to illustrate this, and clarify or discuss the potential impact arising from the dismatch of station location.

Response: Thanks for this valuable comment.

First of all, the comparison analysis between Aeolus and WPR is based on Youyang station. According to studies of Zhang et al. (2016) and Guo et al. (2021), Shapingba Station was excluded

for comparison, because its distances to adjacent tracks of Aeolus exceeded 1°. The detail description was given in Line 146-149 in the revised manuscript.

Secondly, for the description of WPR at Youyang, we have followed the suggestion and added some detail description in Section 2.1.1 Ground-based wind profile data. The WPR in Youyang share same temporal and spatial vertical resolution of 5 min and 120 m with Shapingba. And the distance of Youyang from radiosonde site in Shapingba is more than 360 km. Please see Line 111-120 in the revised manuscript.

Finally, we have added Figure 1 to illustrate the locations of both ground-based stations and the Aeolus tracks, with Figure 1 in the revised manuscript. The discussion about the potential impact arising from the dismatch of station location was given in Line 146-152 of the manuscript.

2. Section 2.1.1 and 2.1.2 can be merged, and "Shapingba WPR is located at the same station as RS; therefore, the data verification of WPR wind observations was conducted based on RS data in this study." can be incorporated to the original section 2.1.2.

Response: We are appreciated for this comment and have followed this suggestion. The original Section 2.1.1 and 2.1.2 have been merged to be Section 2.1.1 Ground-based wind profile data. The description in Section 2.1.1 in the revised manuscript has also been rephrased.

3. Figures: The X-axis and Y-axis scale in Figure 2 can be shortened to better show the details of scatters. For example, both axis can be adjusted to -20 to 40 m/s. Figures 2–5: only major ticks in both axes are shown. It is inappropriate for a high-quality figure not to show the minor ticks.

Response: We are thankful for the careful comment. Following this suggestion, we have adjusted the axis in original Figure 2 and added minor ticks in other figures. Please see Figure 3-8 in the revised manuscript.

**Minor comments:**

Line 12: "vertical wind" can be expanded to "vertical wind profile"

Response: Thanks for this kind remind. We have modified "vertical wind" to "vertical wind profile" in Line 12 of the revised manuscript.

Line 19-20: "Their root-mean-square deviation increased with height but decreased by 3–4 km." is not clear to me. For example, grammar error exists in "decreased by 3–4 km." Besides, I cannot find any figures in this manuscript can support this conclusion, and should be rephrased.

Response: We feel sorry for the unclear expression, and have rephrased this sentence as "Their root-mean-square deviation increased with height, but decreased at heights between 3 and 4 km."

This conclusion is derived from Figure 6(a) in the revised manuscript, with the blue line illustrating vertical variation of RMSE between RS and the WPR. Please see the Line 19-20 for the rephrased sentence, Figure 6 and the corresponding description in Line 276-279 of the revised manuscript.

Line 27: "between ± 5 m/s" can be revised to "within ± 5 m/s" or "between -5 m/s to 5 m/s"

Response: Thanks for this kind comment. We have revised the expression as "within ± 5 m/s" in Line 27 of revised manuscript.

Line 28-29: Can you please clarify the specific characteristics in "the mean differences"?? Otherwise, the sentence "the mean differences… below 1.5 km" makes nonsense.

Response: We are sorry for the lack of clarity and have rephrased the expression as "large mean differences of both Rayleigh-clean and Mie-cloudy winds versus WPR winds appeared below 1.5km". Please see Line 28-29 in the revised manuscript.

Line 30: ", such that" can be modified to ". In this case" or similar expression.

Response: Thank you for the comment. We have followed it in Line 30-31 of the revised manuscript.

Line 31: "large mean differences of 4-8 km" can be rephrased to "Larger mean differences at the height range between 4 to 8 km".

Response: We are thankful for this comment, and have rephrased the sentence following the suggestion in Line 32 of the revised manuscript.

Line 43-44: "to study" -> "for studying…and predicting extreme weather."

Response: Thanks for the careful comment. We have modified "to study" to "for studying" in Line 43-44 of the revised manuscript.

Line 33: I wonder the logic behind "influenced by the topography of the Tibetan Plateau." Chongqing (the elevation is less than 4000 m) is in the southwest China, lying far away from the Tibetan Plateau (TP). Besides, do you have any evidence for the connection between cloud liquid water in the middle troposphere and the topography of Chongqing (not TP)? If not, this conjecture can be deleted.

Response: We are appreciated for the rigorous suggestion and have deleted this expression in Line 33-34 of the revised manuscript.

Line 84: "and three-dimensional spatial structure" can be removed.

Response: We are thankful for the suggestion and have followed it in Line 84 of the revised

manuscript.

Line 103: How about the accuracy of the RS measurement in Chongqing, or China? The authors may refer to the following important papers:
https://doi.org/10.1007/s00376-010-9170-8
https://doi.org/10.5194/acp-16-13309-2016
https://doi.org/10.5194/acp-21-17079-2021
Response: Thank you for the recommendations. Referring to these papers, we have added some description about accuracy of RS measurement in China as:
"Shapingba station belonged to the network of the L-band sounding system by China Meteorological Administration. The operational radiosonde stations in China widely use GTS1 digital radiosonde as key components of L-band sounding system, which have high accuracy within the troposphere in detecting fine resolution profiles of meteorological factors (Bian et al., 2011; Guo et al., 2016; Guo et al.,2021)."
The corresponding references were also added in Reference. Please see Line 106-110, Line 459-461 and 483-491 in the revised manuscript.

Line 104: "Station Shapingba" -> "Shapingba"
Response: Thanks for the careful suggestion. We have made modification in Line 103 of the revised manuscript.

Line 156 and 174: "Where" should be revised to "where" and no indent before "where"
Response: We are sorry for the format errors and have corrected this kind of errors in Line 160, 168 and 187 of the revised manuscript.

Line 213: "which drift more than 10 kilometers away from the releasing station", the drifting distrance of RS balloone depends on the altitude, and this expression can be revised by referring to Figure 2 in Zeng et al. JGR 2019 (https:// doi.org/10.1029/2018JD029109)
Response: Thanks for the recommendation. We have modified the expression referring to Zeng et al. (2019) as "which can respectively drift as far as 0-90, 2-25 and <10km at 200, 500 and 850hPa away from the releasing station (Zeng et al., 2019)". Please see in Line 233-234 and 549-552 in the revised manuscript.

Line 67: Is the publication year in "Zhang et al. 2017" supposed to be 2015?
Response: Thanks for the kind remind. We have checked the publication year of the reference and correct it in Line 64 of the revised manuscript.

---

## Author Response (AR2)

Dear Editors and Reviewers:

Thank you very much for your careful review and valuable suggestions regarding our manuscript titled "Comparisons and quality control of wind observations in a mountainous city using wind profile radar and the Aeolus satellite" (Manuscript Number: amt-2023-152). We appreciate the time and effort you have dedicated to reviewing our work. We have thoroughly studied your comments and have made the necessary changes to improve the manuscript. To provide a clear overview of the revisions, we have summarized the main corrections in the manuscript and provided specific responses to each of the reviewer's comments. You can find these revisions in the attached file "amt-2023-152-manuscript-version4.pdf" . We hope that these revisions have strengthened the manuscript and addressed the concerns raised by the reviewers.

Reviewer #1:

**General comments**

Wind profile observations are essential to atmospheric science, provides valuable information about the vertical structure of atmosphere and the movement of air masses. This study addressed the limited availability and low resolution of ground-based vertical wind observations in Chongqing, a typical mountainous city in China. To overcome this issue, the authors matched the Aeolus satellite track with ground-based vertical wind observation sites in Chongqing, with the year 2021 as an example. They then conducted verification and quality control studies on wind observations obtained from wind profile radar by comparing them with radiosonde data and the Aeolus satellite's Mie-cloudy and Rayleigh-clear wind products. Overall, this paper provides valuable insights into the comprehensive applications of multi-source wind profile data in mountainous cities with sparse ground-based wind observations. The findings contribute to improving the understanding of vertical wind profiles and emphasize the importance of quality control in vertical wind data analysis. This study is meaningful and I recommend this manuscript for publication in AMT after some revision:

Response: Thank you for the valuable and affirmative feedback and suggestions of our manuscript. We will address each of your points in order:

1. Figure 2 illustrates the procedures of data matching and validation, which utilizes several abbreviations. As figures are independent of the text, please provide the meanings of these abbreviations either within the figure itself or its caption for the convenience for reading.

Additionally, please confirm whether it is RWD or WPR.

Response: We appreciate the reviewer for the careful comment and have updated the caption of Figure 2 to provide the meanings of the abbreviations used. The revised caption now reads as follows: "Flowchart of the multi-source wind profile data matching and verification procedures. WPR stands for wind profile radar, RS stands for radiosonde, EOF stands for empirical orthogonal function construction, and HLOS stands for horizontal line-of-sight." Additionally, we feel sorry for the mistake and have revised "RWD" to "WPR" in the figure. Please see Figure 2 in the revised manuscript.

2. In 2.2.4, the Aeolus HLOS wind products, including the Mie-cloudy and Rayleigh-clear wind products, are conducted quality control with two parameters derived from the Level 2B data products. However, I do not find any information about the number of excluded data and effect of this quality control on the overall datasets.

Response: Thank you for bringing this to our attention. Using the parameters valid_flag and hlos_estimate_error, 18241 Mie-cloudy wind profile samples and 1010 Rayleigh-clear samples were excluded. Through the quality control process, significant reductions in the estimated error were achieved for the Mie-cloudy wind products, from 42.22 m/s to 3.50 m/s. Similarly, for the Rayleigh-clear wind products, the estimated error has been reduced from 78.69 m/s to 4.58 m/s. We have incorporated these information in Line 224-229 of the revised manuscript.

3. In 3.1, This manuscript did not provide specific information regarding the missing data rate in the wind profile radar and radiosonde data used in the study. It is important to note that missing data commonly exist in ground-based observations due to various factors such as equipment malfunction, technical issues, or weather conditions. It is recommended that the authors provide relevant information regarding the missing data rate and discuss how they handled the missing data in their analysis.

Response: We are sorry for this neglect and have explicitly described these information in the revised manuscript. Specifically, the missing data rates are 22.78% for Shapingba WPR, 30.08% for Wulong WPR, and 13.55% for radiosonde data. Different approaches were utilized to address the missing data based on the nature of missing value. For cases where certain levels within a profile have missing values, linear interpolation is employed to fill in the missing data. On the other hand, if an entire layer of data is missing, the entire profile is removed from the analysis.

Please see Line 237-243 in the revised manuscript for details.

4. In Table 1, row 3 and row 2 are not aligned, please check.

Response: Thank you for addressing the formatting issue in Table 1. We have carefully checked and aligned all rows in Table 1 to ensure proper formatting and alignment in the revised manuscript.

5. The descriptions of Figure 5 in Line 271-286 are mainly qualitative description. The readers may want some specific data, such as the proportion of deviations concentrated between -10 to +10 m/s at various vertical levels, which could make the description more specific and objective.

Response: We appreciate your comment on revising the descriptions of Figure 5. We have annotated the proportion of deviations concentrated between -10 to +10 m/s at different vertical levels in Figure 5, and have added relevant description in the figure's caption. In the description of Figure 5, we have provided specific and objective analysis based on these information. Comparing RS with the original WPR data, 98.2% of the deviations distributed within the -10 to +10 m/s range near the surface. However, this proportion decreases with increasing altitude, with only 75.6% of the deviations falling within this range between 6-7 km. Furthermore, when comparing RS with the WPR data corrected using Gaussian filtering and EOF reconstruction, a higher proportion of deviations was observed to concentrate between -10 to +10 m/s at different altitudes. Specifically, the deviations between RS and EOFc WPR exhibit a higher proportion of deviations within the -10 to +10 m/s range compared to those between RS and GF data. Please see Figure 5 and Line 288-295 for the description in the revised manuscript.

6. Line 318-319, "The obtained results .... described in Section 2", what is the obtained results? The author may want to say "the match procedure results", please rephrase this sentence.

Response: We feel sorry for any confusion caused and have rephrase this sentence in Line 336 of the manuscript.

7. Line 321-323, "The PDD ... generally present as ...., with 82.9% of deviations concentrated ....", should be "The PDDs .... generally present as ..., with 82.9% of deviations concentrating ...", please check the sentence.

Response: Thank you for addressing the correction in the sentence. We have corrected the sentence to say "The PDDs... generally present as..., with 82.9% of deviations concentrating..." to ensure proper grammar and clarity in Line 340 of the manuscript.

8. In Line 351-352, Line 362, what does the abbreviation "PWR" mean? I can not find its specific meaning in the previous text. Based on the context in line 56, the author may intend to refer to wind profile radar with abbreviation of "WPR" here. Please use the consistent abbreviation throughout manuscript.

Response: We feel sorry for the incorrect use of abbreviation. You are correct that the abbreviation "PWR" was not defined in the previous text. We have revised the manuscript to consistently use the abbreviation "WPR" to refer to the wind profile radar, ensuring consistency throughout the document. Please see Line 369-370 and 380 of the manuscript.

9. In Line 400, "Following screened by ..." should be "Screened by...".

Response: We are appreciated with the careful comment and have corrected the sentence to say "Screened by..." instead of "Following screened by..." for improved clarity in Line 418 of the manuscript.

10. Line 412-413, please unify the tense of the entire sentence.

Response: Thank you for pointing out this grammar error. We have thoroughly reviewed the tense issue in the sentence and made the correction by changing "exist" to "existed" to ensure consistency throughout the sentence. Please see Line 429-430 of the revised manuscript.

Reviewer #2:

This study addresses the challenges of obtaining wind profile data in a mountainous city in China and demonstrates the importance of utilizing various data sources, including wind profile radar, radiosonde, and satellite-based Aeolus measurements. By matching Aeolus track data with ground-based observations and conducting comparisons with WPR and RS data, the authors provide insights into the correlation and discrepancies among these different measurement techniques. The paper showcases the significance of quality control techniques, such as Gaussian filtering and empirical orthogonal function construction, in improving the accuracy and reliability of wind observations. The analysis reveals the impact of these techniques on retaining wind characteristics and reducing deviations of WPR observations. Furthermore, the study investigates the differences between Aeolus Mie-cloudy and Rayleigh-clear wind products compared to WPR data. By considering factors such as boundary layer dynamics and cloud liquid water content, the authors shed light on the vertical distribution and discrepancies observed at different altitudes.

Overall, this paper highlights the importance of integrating and validating data from multiple sources in atmospheric measurement technology. The findings contribute to a better understanding of wind profiles in mountainous regions, where traditional ground-based observations are limited. The methodology and results presented in this study can guide future research and applications in similar geographical settings, ultimately advancing the field of atmospheric measurement technology and improving our understanding of complex wind patterns. This paper is worthy of publication for AMT with the following comments considered.

Response: We appreciate your efforts in reviewing our manuscript and providing valuable comments. Point-to-point response have been summarized as below:

1. The data acquisition rate, which represents the ratio of the actual obtained valid data quantity to the total expected data quantity, is an important quantitative indicator for assessing the detection capability of wind profile radars. It is suggested that the authors provide information about the data acquisition rate for the two radars of Chongqing in Section 2.1.1 to enable readers to better understand the detection capability of WPR in mountainous cities.

Response: Thank you for your suggestion. In the revised manuscript, we have provided information of missing data rates of both WPRs in Chongqing. The missing data rate can complement the data acquisition rate in characterizing the WPR's detection capability in mountainous cities. Please see Line 237-239 of the revised manuscript for the missing data rates of WPRs.

2. In Section 2.1, wind profiler radar, radiosonde, and Aeolus satellite data are described. As far as we know, these data are in different special and temporal resolutions. Therefore, it is necessary to provide information in the manuscript regarding the data quantities obtained from each measurement technique. This will facilitate a better understanding of the data-matching methods and enhance the credibility of the research.

Response: Thanks for this valuable comment. We have included information in the revised manuscript about data quantities of obtained from each measurement technique. Please see Line 224-227 for Aeolus, Line 237-239 for the description of WPR and RS .

3. In Line 60, the reference "Dibbern et al., 2001" is not found in the Reference part. Besides, the omission of reference for "WMO, 2001" in Lines 73-74 also occurred.

Response: We feel sorry for the neglect and have included the references in the Reference part.

The detail of "Dibbern et al., 2001" is:

Dibbern, J., W. Monna, J. Nash, and G. Peters. 2001. COST Action 76-final report. Development of VHF/UHF wind profilers and vertical sounders for use in European observing systems. European Commission, 350 pp.

While the detail of "WMO,2001" is as below:

World Meteorological Organisation (WMO). 2001. Statement of Guidance Regarding How Well Satellite Capabilities Meet WMO User Requirements in Several Application Areas, WMO Satellite Reports SAT-26, WMO/TD No.1052.

Please see Line 487-489 and 565-567 of the revised manuscript.

4. In Lines 75-76, the citation of "Baker et al., 2008" should be "Baker, 2008", because there is only one author for this paper

Response: Thank you for pointing out the error. We have made correction following your suggestion in Line 75-76 of the revised manuscript.

5. In Lines 142-144, the expression in the sentence is strange, and "considering" may be revised to "to make up". Please check and rephrase this sentence to make it more accurate and clear.

Response: We appreciate your suggestion on the expression and have changed "considering" to "to make up" as you suggested in Line 142-144 of the revised manuscript.

6. In Line 233-234, spaces are missing between the numerical values and their units, like "10km" and "850hPa", please review the manuscript for this error.

Response: Thank you for pointing out this clerical error. We have reviewed the manuscript and correct this error. Please see Line 244 and 257 in the revised manuscript.

7. In Lines 254-255, "quality control 1" should be "Quality Control 1" or "the initial quality control". Besides, I guess "WRP" at the end of the sentence should be "WPR".

Response: We appreciate your careful comment and have made correction from "quality control 1" to "Quality Control 1". Besides, "WRP" at the end of the sentence have been modified as "WPR". Please see Line 265-266 of the revised manuscript.

8. Lines 276-277, in the sentence "large positive deviations were mainly distributed around 3-5 km", "were" should be deleted.

Response: Thank you for identifying the error. We have removed the word "were" from the sentence to improve its clarity in Line 288 of the revised manuscript.

9. In Figure 5, I wonder whether the word "RWP" in the legend should be "WPR" to match the abbreviations in the title of the figure.

Response: We appreciate your observation regarding Figure 5. We have updated Figure 5 and changed "RWP" in the legend to "WPR" in the revised manuscript.

10. In Lines 467-469, the publication year for the reference is not found, please check and make revisions.

Response: We are sorry for the neglect and have added the publication year for the reference in Line 484-486 of the manuscript.

Sincerely,

Authors

---

## Author Response (AR3)

Dear Editors:

Thank you very much for your careful review and valuable suggestions regarding our manuscript titled "Comparisons and quality control of wind observations in a mountainous city using wind profile radar and the Aeolus satellite" (Manuscript Number: amt-2023-152).

Suggestion by the handling associate editor:

A minor point may be that the authors use scatter plots with many data points, were most points are actually not visible, since they are overplotted by other points. In such cases it is better to provide a measure of data density in tghe plot, which could be done by color or contours. In case two scatter plots are compared, one could be in color, overplotted by data density contours of the other. In that case all points are represented in the plot.

Response: Thanks for this valuable suggestion. We have replaced the scatter plots in Figure 3(a)-(b), Figure 4 with data density contour plots and Figure 7 with probability density distribution plots, and updated the corresponding descriptions of these figures. This change was made to address the issue of point overplotting and provide a clearer representation of data distribution. Please see Figure 3, 4 and 7 , Line 268-274, Line 343, 353-355 and Line 360-362 in the revised manuscript.

Sincerely,

Authors